# Innovative Collaboration between a Medical Clinic and a Community Pharmacy: A Case Report

**DOI:** 10.3390/pharmacy7020062

**Published:** 2019-06-14

**Authors:** William R. Doucette

**Affiliations:** College of Pharmacy, University of Iowa, Iowa City, IA 52242, USA; william-doucette@uiowa.edu

**Keywords:** comprehensive medication review, community pharmacy, collaboration

## Abstract

As value-based payments become more common in healthcare, providers can develop collaborative relationships to support performance. A medical clinic and community pharmacy worked together to deliver collaborative medication management services to targeted patients in an accountable care organization. The community pharmacy was paid by the clinic to conduct comprehensive medication reviews (CMRs) for 116 patients. The CMRs initially were delivered to patients taking at least 10 medications and to patients rated as high cost/risk by the clinic. The most common medication-related problem types were Needs additional therapy (38.8%) and Suboptimal therapy (19.0%). The most common pharmacist actions were to Change medication (18.1%) and Initiate new therapy (13.8%). Financial analyses showed net savings in annual patient out-of-pocket expenses just over $15,000 for the cohort of patients, and net annual direct cost savings from a payer perspective of about $70,000. This innovative partnership between a medical clinic system and a regional pharmacy chain built upon initial discussions and planning. The partners were able to address problems that arose with their collaboration, changing their approach as needed. The outcomes were positive for the clinic and pharmacy, their patients and the payer(s). Interested providers are encouraged to pursue similar collaborations, which could be key to success in today’s healthcare environment.

## 1. Introduction

The current healthcare environment has made value-based payments a popular approach [1,2]. Under such programs, providers often can benefit from closer coordination with each other across the continuum of care. An important provider dyad, especially for managing chronic conditions, is the medical clinic–community pharmacy. Both of these providers are involved with helping patients achieve positive outcomes from medication therapy while working to limit total costs of care [2,3]. 

Though some clinic–pharmacy coordination is occurring, few detailed descriptions have been published about successful collaboration between clinics and community pharmacies under the evolving value-based payment environment [4,5]. Over the past five years, the Iowa Department of Public Health has worked with a team at the University of Iowa College of Pharmacy to foster team care among clinic–community pharmacy partners to improve management of hypertension and diabetes. This work has stimulated some clinic–community pharmacy collaboration. The objective of this case report is to describe the development, operations, and outcomes of an innovative clinic–community pharmacy collaborative care model for providers facing value-based payments.

McFarland Clinic has been a leading multispecialty clinic organization in Iowa for over 70 years. It is Iowa’s largest physician-owned multispecialty clinic. The McFarland Clinic provider network serves patients through about 20 locations in central Iowa, delivering over 1 million patient visits annually, involving more than 280 providers and 1000 staff members. McFarland Clinic embraces innovation which enables excellence in clinical practice. Given the presence of value-based payment programs by accountable care organizations (ACOs), McFarland Clinic leaders were considering ways to successfully apply a population health management model to support their value-based performance.

NuCara Pharmacy is a full-service regional pharmacy chain with more than 20 locations in four states. NuCara strives to provide the best products, the best service, and latest innovations to benefit their patients and customers. Their pharmacists work with patients and other providers to optimize medication therapy, in addition to delivering a range of pharmacy services. Their services include compounding custom prescription medications, medication management, immunizations, medication synchronization, medication packaging, infusion services, and sales of home medical equipment. They participate in a value-based pharmacy program operated by a large payer, which made closer coordination with clinics of interest to them. 

## 2. Materials and Methods

To gather data about this case study, a key informant from both the participating clinic and the participating pharmacy was interviewed, using semi-structured questions. The set of interview questions was developed from an initial description of the clinic-community pharmacy joint activities by a participating pharmacist. The questions asked about having the pharmacy conduct comprehensive medication reviews for patients using a lot of medications, a pharmacy medication management service for high risk/high cost patients, and the start of medication adherence management service provided by the pharmacy. The interviews were audiotaped, transcribed verbatim, and then analyzed. The responses were coded for the topics describing how the partners developed their joint activities, as well as how they operated. In addition, some documents describing the partnership were shared with the investigator.

In addition, a financial analysis was conducted by the pharmacy to estimate the net benefit associated with the joint activities. A conservative approach was followed, focusing on cost savings resulting from changes in medication therapy made in response to the CMRs, primarily from brand or non-formulary medications to generic formulary medications. Such savings were estimated for 12 months, from a payer perspective and for patient out-of-pocket expenses. Medication costs were calculated using insurance payment formulas for the insurance the patients had, or a cash formula used for that medication. Estimated actual acquisition costs for the pharmacy were used for product costs. No related cost estimates for reduced clinic visits, ER visits or hospitalizations were included in the financial analysis.

## 3. Results

Initial Teamwork: Both of these providers have locations in Ames, Iowa that participated in an initial effort at team management of patients with diabetes, working with a team at the University of Iowa. For this first effort, representatives of the McFarland Clinic and NuCara Pharmacy met with personnel from the University of Iowa and the Iowa Department of Public Health to discuss how the team approach could work between their organizations. Though these providers worked to collaborate, this initial effort was limited by having few targeted patients not meeting diabetes goals (e.g., hemoglobin a1c target, BP target). Thus, this coordinated work on team management of medications for patients with diabetes did not persist. However, it did lay the groundwork for later collaboration, as the providers were able to better understand each other’s practices, appreciate their respective expertise and establish a working rapport.

Medication Management for High Medication Use Patients: One approach pursued by the clinic to try to improve ACO performance metrics was to optimize medication therapy for their patients who were taking at least ten different medications and were covered by an insurer’s ACO program. The clinic approached NuCara Pharmacy in Ames about providing a comprehensive medication review (CMR) service for a panel of patients with high medication use. The primary discussions occurred between the physician and pharmacist who had worked together in the initial team management for patients with diabetes. After a mutually agreeable plan had been outlined, each provider worked with their own organization’s leadership to get support to proceed. The pharmacy was willing to provide a comprehensive medication management service, for which the clinic paid the pharmacy using a fee for service approach. The physician leader talked with the clinic’s providers about the program to gain their support and feedback. No formal agreement was signed initially, to allow flexibility to adjust the collaboration, though one was signed later.

To begin the service, the clinic identified a list of about 200 patients in the Ames area to receive a comprehensive medication review (CMR) by one of the pharmacy’s pharmacists. These patients were taking at least ten different medications, which was considered high medication use. Some of these patients got their medications dispensed from the participating pharmacy, while most did not. The clinic sent an initial letter to each patient that described the program and encouraged them to work with the pharmacy to receive a CMR. After the letters were sent out, the pharmacy phoned the patients on the list to determine their willingness to participate in the program and to schedule CMRs for interested patients. There was a low response rate from patients initially. To address this issue, the clinic revised the patient letter and added the patient’s provider’s signature. Also, the NuCara pharmacist tweaked the telephone contact script. Together, these changes improved the patient participation in this CMR program. The pharmacy provided collaborative comprehensive medication management services to 85 patients in this first phase.

Most of the CMRs were conducted over the telephone, though some were completed in the pharmacy. The pharmacist first gathered patient information from the available patient records (either from the pharmacy or from the clinic EMR) to get an initial view of the patient’s medication regimen and labs. Then, the pharmacist talked with the patient to collect additional information about how the medications were being taken, and if likely adverse effects were present. While on the call with the patient the pharmacist discussed a medication plan to address any identified problems. The plan was implemented by working with the patient and/or the involved provider(s). Communication with providers used multiple modes, including secure text through the EMR, faxing the clinic, and phoning the clinic.

Another obstacle was limited responses by clinic providers to clinical notes sent by the pharmacist. It was typical that the pharmacist identified either a question or a potential medication-related problem when reviewing a patient’s medications. In such instances the pharmacist sent a note to the involved provider(s) asking them to clarify the situation or to adjust medication therapy. Unfortunately, since this was the first time many of the clinic providers were collaborating closely with a community pharmacist on medication management, the providers hesitated to respond to pharmacist notes/queries. The pharmacist and physician leader discussed this situation, and the physician leader communicated with the clinic providers to clarify the objectives of the collaboration, asking them to be responsive to the pharmacist’s notes/queries.

The pharmacist worked with the patients’ dispensing pharmacies to get their medication lists and/or dispensing records, which were used for the CMRs. It required considerable effort to get other pharmacies to provide medication information, but some did eventually provide patient medication information requested by the pharmacist. To get more complete clinical information about the listed patients, the pharmacist worked with the clinic to get access to the clinic’s EMR. The EMR provided the pharmacist with clinical information that allowed more efficient and effective CMRs to be conducted. In addition, the pharmacist was able to use a secure texting feature in the EMR. However, its effectiveness in communicating with the clinic providers was variable because not all of them used it.

Medication Management for High Risk Patients: After the success of the initial teamwork to address patients with high medication use, the clinic and pharmacy developed another joint effort focused on patients with high risk and/or high cost. From a list of high risk patients from the clinic, 31 patients received comprehensive medication review services from the pharmacy in the second phase of this collaboration. The clinic continued to pay the pharmacy to provide the collaborative CMR service. As with the previous teamwork, the clinic notified the patients that the pharmacy would be contacting them about the CMR service. The same collaborative CMR service was delivered to this second group of patients, who typically had multiple chronic conditions or were taking high cost medications. As with the first group, not many of these patients were getting their medications dispensed at NuCara Pharmacy. Though the communication processes were smoother than during the initial phase, some patients declined to have the CMR service. Similarly, the responsiveness of providers was somewhat better with this second group, after having the processes improved during the work with the first group of patients.

The partners conducted an aggregate evaluation of the collaborative comprehensive medication review services for the two groups. This work provided CMRs for a total of 116 patients. One component of their assessment was to characterize the medication-related problems identified by the pharmacist when conducting the CMRs. The most frequent types of medication-related problems were “Needs additional therapy” (38.8% of patients) and “Suboptimal therapy” (19.0%) (Appendix A).

In addition, the pharmacist’s actions were summarized. The most common pharmacist action was to work with the provider and patient to change a medication (18.1%) (Appendix A). The pharmacist also worked to initiate new drug therapy (13.8%) and helped to discontinue unnecessary or problematic medications (12.9%). The pharmacist also educated patients to improve their medication administration technique (e.g., use of inhaler) (11.2%). 

For the financial analysis, the net cost savings associated with pharmacist services under the partnership were calculated from a patient perspective and a payer perspective. The estimated 12-month direct cost savings for the 116 patients totaled $15,483. Also, direct cost savings on medications for a payer were estimated to be $70,513.74 over 12 months. Based on a variable fee amount for the different patient groups, the cost to the clinic for paying the pharmacist to conduct the CMRs totaled $13,150.

Current State of Collaborative Medication Management: The partners have built on the success of the comprehensive medication review service by implementing a service targeted at improving medication adherence, which is a common challenge for chronic medication therapy. This step was an initiative raised by the pharmacy to work together to improve medication adherence for shared patients: Adherence Services Referral Program. The adherence program can help the pharmacy perform well on metrics in a value-based pharmacy program in which it is participating. For this adherence program NuCara Pharmacy worked together with Medicap, another regional pharmacy chain who also has a pharmacy in Ames and others in central Iowa. Together these pharmacies are working with McFarland Clinics in their communities to identify and address medication nonadherence for only their own dispensing patients. Building on the success of the collaborative CMRs with one NuCara Pharmacy, McFarland Clinics is working with over a dozen pharmacies between NuCara and Medicap pharmacies–with all of the pharmacies participating in a value-based pharmacy program that includes adherence-based metrics. A progressive aspect of this newer program is that McFarland has allowed these pharmacy partners to access their EMRs through the web-based EpicCare Links. Such a connection creates efficiencies for the pharmacists caring for mutual patients of McFarland Clinics, while eliminating some phone calls to the clinic. Though it is too early to evaluate this adherence program, it is promising.

## 4. Discussion

This case report describes collaboration that occurred primarily over about 18 months between McFarland Clinic and NuCara Pharmacy. These partners built on an initial team-based project to implement collaborative comprehensive medication review services, delivered by a pharmacist working with clinic providers and patients. There were several innovative components of this collaboration, including: a clinic paying a community pharmacy to improve patient care under an ACO model, the clinic actively engaging patients and providers to participate with the pharmacy services, and the clinic connecting with the pharmacy via its EMR. Given the widespread presence of ACOs, and the many clinics that do not employ pharmacists, this model could be useful to a large number of providers. Progressive clinics and pharmacies that want to succeed under the value-based payment programs being used more commonly today should explore the feasibility of implementing a similar approach in their communities [6].

Facilitators and obstacles of this collaborative approach can be identified. An important facilitator is having a trusted, motivated partner [7,8,9]. In this case, the partners gained familiarity by working on a small project facilitated by the Iowa Department of Public Health and the University of Iowa. Other prospective partners can identify opportunities to work together to address a problem in their community or among mutual patients. Perhaps it is finding a way to improve medication adherence, or in coordinating to better manage a key chronic condition, such as diabetes or hypertension, represented in performance metrics. Working together will give the partners an opportunity to get to know each other better, which is a starting point for developing a working rapport and trust [10].

A related facilitator for establishing successful clinic-pharmacy collaboration is to start small [10]. Both of the key practitioners from the clinic and pharmacy partners in this case identified starting in a limited way as an important factor in their progress. The clinic approached the pharmacy about providing some type of service that could help with their patients who were the highest medication users. The pharmacy had some experience with providing comprehensive medication reviews, and offered that as a viable option. Upon discussion, the partners agreed that the clinic would send a list of patients using at least ten medications, who became the targets for the first phase of the medication review service. This relatively narrow focus allowed the clinic and pharmacy to work through issues that arose as the new program was implemented, including communications with patients, responsiveness of clinic providers to pharmacist notes and pharmacy access to the clinic EMR. Potential partners seeking to pursue a similar collaborative model would benefit from discussing patient groups or issues to get their collaboration started. Then, spend time clarifying roles of the parties involved, including what services are to be performed, how communication will be conducted, and details for any payments or revenue sharing. Starting small, but with a clear plan, should support positive results on innovative collaboration for pharmacy services.

A third factor that can contribute to a successful collaborative working relationship is to collect data that track performance on key variables or metrics [6]. In this case, the parties tracked the number and type of medication-related problems identified and managed through the comprehensive medication review service. In addition, they examined finances associated with the program. It takes some planning and effort to determine which data to collect, how it will be collected, and then how it will be analyzed. Each of these partners had some resources and experience in evaluating new services or programs, which helped them make a feasible evaluation plan for their collaboration. In addition, they worked with Telligen, the CMS quality improvement organization (QIO) that serves Iowa. While Telligen did not have data for the commercial ACO, they did have data for the small number of Medicare beneficiaries who received a CMR. Partners should prospectively plan for some type of evaluation of their efforts at collaboration. Such an assessment can be used internally to build support for the program (e.g., among providers), as well as externally (e.g., to talk with payers or other potential partners).

An obstacle encountered by these partners was limited responsiveness by clinic providers to pharmacist notes sent either by fax or via the clinic EMR. This likely occurred since many of the providers were not used to receiving clinical notes from pharmacists. Rather, their communications with pharmacists tended to focus on renewing prescription orders or clarifying questions on new orders. For the CMR service, the pharmacist had to communicate with the providers about potential problems with the medications, sometimes with suggested changes in therapy. Again, the clinic providers may not have wanted to receive such feedback at first and ignored it. The lead clinic physician spent more time communicating with his colleagues about the collaboration, and the responsiveness of the providers improved. Prospective partners, especially within a clinic or system, should have discussions beforehand about what types of communications they could receive and how they should respond to them. Having clear expectations about communication should smooth out some of the progress when implementing a new collaborative program.

Patient acceptance was a challenge for the collaborative comprehensive medication review service delivered to the two patients groups identified at the clinic. This mostly arose because not all of the patients were regular patients at the partner pharmacy. Rather, the pharmacy was serving as the provider for the CMR services for the clinic’s patients. Though the clinic sent letters to the patients before the pharmacy telephoned them, over half of the patients initially declined the CMR service. An adjustment in the letter from the clinic, incorporating their physician’s agreement, helped the providers and patients both have more confidence in that particular patient receiving the CMR service. The pharmacy also made helpful changes in their phone script when contacting the patients. The latest service, focused on medication adherence, is directed only at current patients of the clinic and pharmacy. Thus, the difficulty of not having a relationship with the patient will not be present for the pharmacy. Prospective partners can address this issue by limiting their collaboration to only mutual patients. In addition, having an announcement about any collaborative program initiated by the clinic would be vital to informing patients about a new service opportunity.

## 5. Conclusions

In conclusion, this case report describes an innovative partnership between a medical clinic system and a regional pharmacy chain. Both organizations are innovative, and were willing to work together. Their collaboration built upon initial discussions and planning. They were able to continue to communicate to address problems that arose with their collaborative program. The outcomes were positive for both the clinic and pharmacy, their patients and the payer(s). The clinic improved performance associated with ACO metrics and received helpful clinical input from a pharmacist. The pharmacy expanded its services and generated new revenue. Patients received more coordinated care, which supported some cost savings for the patients and payers. Interested clinics, health systems and pharmacies are encouraged to pursue similar collaborations, which could be key to success in today’s healthcare environment.

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
