# Peer review of "Innovative Collaboration between a Medical Clinic and a Community Pharmacy: A Case Report"

_pharmacy, 2019, doi:10.3390/pharmacy7020062_

Round 1
Reviewer 1 Report
See attachment for detailed comments.

Author Response
Reviewer 1
1. If you have more details about your financial analysis process, this may be worth including.
Response: We have added some details to the Methods about the financial analysis.
2. Much of the information in the Initial Teamwork, Medication Management for High Medication Use Patients, and Medication Management for High Risk Patients sections seems to be more descriptive in nature about how the collaboration was set up, how patients were identified, etc. so would fit better in the Methods.
Response: Since this is a Case report, the description of the services and associated changes that were conducted by the partners fits best in the Results. If this was a controlled trial or more rigorous study, we agree the suggested changes would be good. However, in this case, we declined to move the service and workflow descriptions from the Results.
3. Abstract, Line 17: consider changing “group” to “cohort of patients”
Response: Suggested change made.
4. Abstract, Line 18: consider removing “in conclusion” at the beginning of the sentence
Response: Suggested change made.
5. Abstract, Line 20: consider adding on to end of sentence “and make appropriate changes if necessary”
Response: Similar change made.
6. Line 59-60: consider alternative sentence wording “The set of questions was based on initial description of the clinic-community pharmacy joint activities with a pharmacist.”
Response: Similar change made.
7. Line 64: consider replacing “coding described” with “responses were coded for the topics describing”
Response: Suggested change made.
8. Lines 64-66: consider switching the order of these 2 sentences
Response: Switched order of sentences as suggested.
9. Line 66: “internal documents” is vague, what was included in these documents?
Response: Text was clarified to better describe the documents.
10. Line 80: consider adding “between their entities” to the end of the sentence
Response: Similar change made.
11. Line 81: consider replacing “at goal” with “meeting diabetes goals”
Response: Suggested change made.
12. Line 91: consider replacing “teamwork” with “team management for patients with diabetes”
Response: Suggested change made.
13. Line 92: consider alternative sentence wording “After a mutually agreeable plan had been outlined, each provider worked within their own organizations to get support to proceed.”
Response: Suggested change made.
14: Line 94: provide more details regarding how the clinic paid the pharmacy, see line 166 comment
Response: Added text to clarify that the clinic paid the pharmacy on a fee for service basis.
15. Lines 94-95: remove sentence “The physician…” as it is repetitive
Response: No change, as this sentence adds to the description of the collaboration.
16. Line 98: consider replacing “sent” with “identified”
Response: Suggested change made.
17. Line 105: consider alternative sentence wording “There was a low response prate from patients initially.”
Response: Suggested change made.
18. Line 108: Consider replacing “to” with “in”
Response: Suggested change made.
19. Line 110: any numbers on the number of CMRs completed in person vs via phone?
Response: No data on mode of CMR were collected.
20. Line 113: consider changing “Then, she” with “The pharmacist the”
Response: Suggested change made.
21. Lines 115-116: consider changing “Then she implemented the plan” with “The plan was implemented”
Response: Suggested change made.
22. Line 120: consider adding “either a medication-related question”
Response: Suggested change made.
23. Lines 119-127: this paragraph may be more of something to address in the Discussion vs the Results section
Response: We considered this change, but kept the paragraph in Results, since we believe it fits best there.
24. Lines 128-136: this paragraph may fit better after the sentence on lines 111-113
Response: We considered this change, but kept the initial paragraph order, which we think has the best flow.
25. Line 140: what constituted a high risk and/or high cost patient for your study?
Response: These were determined by the clinic’s analytics group, and specifics are not available.
26. Line 140: it mentions 31 patients completed the CMR, but what was the total number contacted, this information was provided for the first group of patients so would be good to include for this group as well
Response: We do not have the denominator for this, only the actual number of patients the participated.
27. Line 151: this is where I think the Results section should start, could have 2 subsections, “Medication Interventions” and “Financial Analysis”
Response: We believe that such subheadings are not needed, and have not made this suggested change.
28. Line 166: $13,150 how was this sum determined? standard payment amount per patient, variable payment amount per patient depending on patient specific factors, lump sum as a percentage of total savings?
Response: Clarified that these payments were fee for service totals for all the patients served.
29. Line 174: consider adding “regional pharmacy chain”
Response: Suggested change made.
30. Current State of Collaborative Medication Management section could stay as part of the Results section since it could be considered a “result” of the established collaboration or could be incorporated into the Discussion as well
Response: We kept it in the Results section.
31. Line 203: consider using “hypertension” instead of “high blood pressure”
Response: Suggested change made.
32. Line 209: consider adding “help with their patients who were the highest medication users”
Response: Suggested change made.
33. Line 212: spell out number 10 → “ten”
Response: Suggested change made.
34. Line 216: consider replacing “do well to discuss” with “benefit from discussing”
Response: Suggested change made.
35. Line 208: consider replacing “done” with “conducted”
Response: Suggested change made.
36. Line 219: consider deleting “that could be made”
Response: Suggested change made.
37. Line 226: consider replacing “what will be done once it is available” with “how it will be analyzed”
Response: Suggested change made.
38. Lines 230-231: consider changing wording of beginning of sentence to “Partners should prospectively plan”
Response: Suggested change made.
39. Lines 251-252: how did changing the letter make providers and patients more confident? possibly consider discussing what some of the specific changes that were made instead
Response: Added some text about the changes in the letters.
40: Line 265: expand on the positive outcomes seen
Response: We added text to illustrate the benefits.
41. Line 265: add word “both the clinic”
Response: Suggested change made.
42. Consider moving the N up in the table to the header
Response: Suggested change made in both tables.
43. Line 280: consider just using “Nonadherence” and removing “Under use of medications”
Response: Suggested change made.
Reviewer 2 Report
Overall, this is a well written manuscript that describes the relationship between a large clinic and pharmacy chain to improve patient care through comprehensive medication reviews. The information that sets up the case report is very helpful to understand how the process evolved over time. The manuscript contains a few areas where additional text would improve the information shared with readers.
On page 2 in line 81, it would help the reader by defining what was meant by goal.It is obviously a diabetes goal but it would be helpful to delineate the description of the goal.
On page 3 in the paragraph bound by lines 110 to 118, it is necessary to call out the pharmacist by the pronoun she.
On page 4 in the last paragraph starting at line 162, the manuscript would benefit from additional details about how the costs were determined. This is a critical finding of the manuscript and as written, the reader does not have enough information to know how this determination was made.
In addition, it appears that there are a few places in the manuscript where a comma is needed such as p1 line 34 after years; and p3 line 115 after patient.
Author Response
Reviewer 2
44. On page 2 in line 81, it would help the reader by defining what was meant by goal. It is obviously a diabetes goal, but it would be helpful to delineate the description of the goal.
Response: We have added text to clarify this refers to diabetes goals and included examples.
45. On page 3 in the paragraph bound by lines 110 to 118, it is necessary to call out the pharmacist by the pronoun she.
Response: We changed “she” to “the pharmacist”, following a suggestion by another reviewer.
46. One page 4 in the last paragraph starting at line 162, the manuscript would benefit from additional details about how the costs were determined. This is a critical finding of the manuscript and as written, the reader does not have enough information to know how this determination was made.
Response: We have added details to the Methods more fully describing the financial analyses.
47. In addition, it appears that there are a few places in the manuscript where a comma is needed, such as p.1 line 34 after years; and p.3 line 115 after patients.
Response: The first suggested change was made, but the second one was not as a comma was not needed there.
Round 2
Reviewer 1 Report
Thank you for your comments regarding the suggested changes. The updates that were made positively added to the overall manuscript.